# A SIMULATION-HEURISTICS DUAL-PROCESS MODEL FOR INTUITIVE PHYSICS

## ABSTRACT

The role of mental simulation in human behavior for various physical tasks is widely acknowledged, attributed to the generality of Intuitive Physics Engine (IPE). However, it remains unclear whether mental simulation is consistently employed across scenarios of different simulation costs and where its boundary is. Moreover, cognitive strategies beyond these boundaries have not been thoroughly investigated. Here, we adopted a pouring-marble task containing various conditions to study IPE's limits and strategies beyond. A human study revealed two distinct error patterns in predicting the pouring angle, differentiated by the simulation time using a boundary. This suggests a possible switching of the underlying reasoning strategies. Our initial experiment on IPE showed that its correlation with human judgments diminished in scenarios requiring extended time of simulation. This observation prompted the exploration of an alternative mechanism based on heuristics for intuitive physics. We uncovered that a linear heuristic model, relying exclusively on empirical data, replicated human prediction more accurately when the simulation time exceeded a certain boundary. Motivated by these observations, we propose a new framework, Simulation-Heuristics Model (SHM), which conceptualizes intuitive physics as a dual process: IPE is predominant only in short-time simulation, whereas a heuristics-based approach is applied as IPE's simulation time extends beyond the simulation boundary. The SHM model aligns more precisely with human behavior across various scenarios and demonstrates superior generalization capabilities under different conditions. Crucially, SHM integrates computational methods previously viewed as separate into a unified model, quantitatively studying their switching mechanism.

**Keywords:** intuitive physics; physical reasoning; mental simulation; heuristic model

## 1 INTRODUCTION

Humans demonstrate extraordinary abilities in understanding and reasoning about the physical world even without formal training in physics (Piloto et al., 2022). This ability, known as intuitive physics (Kubricht et al., 2017b), enables comprehending physical concepts (Baillargeon et al., 1985; Baillargeon and Graber, 1987; Kim and Spelke, 1992), predicting physical dynamics (Battaglia et al., 2013; Bates et al., 2015; Davis et al., 2017), and interacting with the physical environments (Allen et al., 2020). However, human intuitive physics may exhibit errors and biases in certain physical scenarios, indicating deviations from classical Newtonian physics (McCloskey et al., 1980; 1983; Kaiser et al., 1986; Kozhevnikov and Hegarty, 2001). Such errors and biases serve as a unique aspect of human reasoning, offering a valuable avenue for studying the underlying mechanisms of intuitive physics (Kubricht et al., 2017b).

A common perspective to understanding human intuitive physics is mental simulation: it hypothesizes an approximate intuitive physics engine in the human mind (Battaglia et al., 2013; Smith and Vul, 2013; Ullman et al., 2017; Smith et al., 2024). This simulation framework, grounded in probabilistic inference, was found to be able to characterize human behavior across various physical tasks, and also account for human errors and biases, further validating its relevance and applicability (Battaglia et al., 2012; Kubricht et al., 2016; 2017a; Gerstenberg et al., 2017; Ullman et al., 2018; Bass et al., 2021; Chen et al., 2023; Li et al., 2023). Nevertheless, the simulation model

Figure 1: **(A) Experimental design:** Trials involved 3 cup shapes (H-shape, A-shape, V-shape), 3 object shapes (circle, triangle, trapezoid), 3 sizes (large, medium, small), and 2 filling heights (full, half), totaling 54 unique conditions. Participants predicted the tilt angle for marbles to fall out when cups are tilted to the left. **(B) SHM hypothesis:** Participants used either mental simulation, simulating the tilting process until pouring out, or a heuristic strategy, reaching judgments from physical features when the simulation exceeds a boundary. These methods could result in different outcomes. **(C) Human results:** Each point represents a condition, illustrating human tendencies to either overestimate or underestimate the pouring angle. The red and blue lines are the regression results of IPE and the heuristic model, respectively. The SHM effectively captures human behavior with a switching boundary.

fails to completely explain the variance in human behavior in some demanding or unfamiliar conditions (Schwartz and Black, 1999; Kozhevnikov and Hegarty, 2001; Smith et al., 2018; Ludwin-Peery et al., 2021), suggesting the existence of alternative cognitive mechanisms, possibly mental shortcuts employed for certain physical scenarios, or heuristics (Kozhevnikov and Hegarty, 2001; Kubricht et al., 2017b; Smith et al., 2018).

Here we ask the following questions: *Do humans consistently rely on mental simulation, or do they employ alternative heuristic strategies under certain conditions? What are the circumstances that prompt a switch between these two cognitive strategies?*

Previous studies have investigated the interplay between simulation and heuristics, providing evidence for **qualitative** insights. For instance, Kozhevnikov and Hegarty (2001) demonstrate that people tend to use impetus heuristics in quick judgment scenarios, while Battaglia et al. (2013) find that models based on height heuristics can more accurately explain human judgment in certain tasks, such as predicting the falling distance of a block tower. Furthermore, Smith et al. (2017) suggest the integration of these two cognitive strategies in a motion prediction task. However, there is currently no study that has provided clear evidence supporting the relationship between these two strategies or **quantitatively** demonstrated the transition between them. A comprehensive exploration is needed to understand whether a switch of policies exists and, if so, how these switches operate, as well as to identify alternative heuristics that could reverse engineer the human physical reasoning process, including human biases.

In our study, we systematically investigate the switch between simulation and heuristic strategies in intuitive physics, developing a computational model that offers improved explanatory power. We hypothesize that: (i) the simulation strategy prevails in scenarios simple enough for reliable physical unfolding; (ii) the heuristic strategy takes over when mental simulation becomes too costly; (iii) the switching point of the two strategies correlates with the simulation cost, approximated via a proxy of simulation time. Diverging from previous studies that often focused on simpler dynamics or predictable outcomes (Battaglia et al., 2013; Smith and Vul, 2013; Smith et al., 2017), our study engages in examining human reasoning across a range of simulation costs (Schwartz and Black, 1999; Kubricht et al., 2016; Davis et al., 2017). Inspired by previous pouring tasks in intuitive physics (Schwartz and Black, 1999; Kubricht et al., 2016; Guevara et al., 2017; Lopez-Guevara et al., 2020), we build a pouring marble task with more diverse physical properties and complexities.

In this task, human participants are asked to judge the tilt angle needed to pour marbles from cups under various setups (see fig. 1A).

We conducted four steps of experiment to validate the above hypotheses sequentially. The **first step** examines whether there is a pattern switch regarding human judgment. A finding of two distinct error patterns (i.e.overestimation and underestimation) supports the existence of two predominant strategies that vary under different simulation times. The **second step** aims to test our hypothesis on whether the IPE model can account for human judgments in simpler scenarios. The results show that it aligns well with human judgments and exhibits the same overestimation when the actual pouring angle is small. However, it fails to account for humans' underestimation as the actual pouring angle exceeds a certain boundary (see fig. 1C). Given that the pouring rate remains consistent, we hypothesize longer simulation time leads to increased cost of physical unfolding, triggering the transition to another cognitive strategy. Thus, we validate our second hypothesis by exploring an alternative heuristic approach in the **third step**. We developed a linear heuristic model trained on ground-truth data and found that, although less effective than IPE at smaller angles, the model accurately captures the underestimation pattern when the pouring angle exceeds a certain boundary. These results support our hypothesis of a cognitive shift to a heuristic strategy. To test our third hypothesis, in the **fourth step**, we explore whether a novel framework, Simulation-Heuristics Model (SHM), that combines these two models and toggles based on simulation cost, can explain human judgments across all complexity levels (see fig. 1B). The results show that SHM aligns more closely with human behavior across diverse scenarios and metrics, enhancing our understanding of intuitive physical reasoning and highlighting the adaptability and versatility of human cognition.

## 2 RELATED WORK

**Multiple systems in intuitive physics** The computational mechanism in intuitive physics has attracted significant attention since Battaglia et al. (2013). While previous studies have provided empirical evidence on the adoption of either a simulation engine or heuristics (Kubricht et al., 2016; Schwartz and Black, 1999), there is still no consensus on how these two strategies can coexist to create a unified understanding of the physical world (Ludwin-Peery et al., 2021; Smith et al., 2023). Currently, research tends to examine these strategies separately by adjusting task settings or stimulus properties. For example, it has been observed that simulation-based reasoning is typically employed when dealing with dynamic and natural stimuli, whereas heuristic reasoning is often used in response to static or abstract stimuli, relying on rule-based shortcuts (Kaiser et al., 1992; Schwartz, 1995; Kozhevnikov and Hegarty, 2001). In our research, we aim to complement this qualitative perspective with a quantitative analysis, particularly focusing on capturing the switch point in strategy selection.

**Tasks in intuitive physics** Research has focused on a variety of physical scenarios: determining the heavier of two objects post-collision (Gilden and Proffitt, 1994; Sanborn et al., 2013; Todd and Warren Jr, 1982), predicting the stability of stacked block towers (Battaglia et al., 2013; Groth et al., 2018; Lerer et al., 2016), assessing whether water will pour at the same angle from different containers (Kubricht et al., 2016; Schwartz and Black, 1999), and understanding the behavior of various materials in dynamic contexts (Kubricht et al., 2017a). However, it has been challenging to determine whether the choice between rules and simulation is a deliberate decision or a fixed response based on the problem at hand. This is because either the available rules or heuristics are significantly less helpful than simulation and are therefore never chosen, or they are overly helpful and are consistently chosen over simulation (Kozhevnikov and Hegarty, 2001; Kubricht et al., 2016; Smith et al., 2017; Kubricht et al., 2017b). In this work, we design multiple controllable variables to create diverse scenarios for human participants to choose their preferred strategies.

## 3 MODELS

### 3.1 MENTAL SIMULATION

Recent work explains human intuitive physics understanding by assuming an approximate simulation engine in the human mind (Battaglia et al., 2013; Lake et al., 2017; Kubricht et al., 2016). This

engine serves to simulate the future physical unfolding, akin to a computational physics engine but incorporates noise into the physical properties of objects.

Following this approach, our model utilizes an IPE that runs noisy simulations as in Battaglia et al. (2013). The model takes an initial physical scene $S_0$ and external forces $f_{0:T-1}$ to derive the judgment $J$. This process involves predicting the intermediate states $S_{1:T}$ over a time span $T$:

$$P(J|S_0, f_{0:T-1}) = \int_{S_{1:T}} P(J|S_{1:T})P(S_{1:T}|S_0, f_{0:T-1})dS_{1:T}, \tag{1}$$

where $S_{t+1} = \phi(S_t + \epsilon, f_t)$ with noise $\epsilon \sim \mathcal{N}(0, \sigma^2)$, and $\phi(\cdot)$ deterministic physical dynamics. We simplify the mapping from the initial state to the final judgment as $M(S_0; f, \epsilon)$.

In our implementation using the flexible physics engine Pymunk, the IPE utilizes all physical variables to simulate future dynamics with added Gaussian noise $\mathcal{N}(0, \sigma^2)$ to each marble's position horizontally and vertically during the simulation process. The noise level $\sigma^2$ is varied from 0.1 to 1 to observe its impact on the simulation results. Additionally, we manipulate the rotational speed of the cup to assess its influence on the outcomes.

We perform 30 noisy IPE simulations per trial. During each simulation, an automatic detection system is integrated to identify the moment when the marbles fall out, which serves as the ground truth. The final pouring angle is determined from the average of the 30 results. This setup allows us to mimic the variability and uncertainty in human cognition, as outlined in prior studies (Smith and Vul, 2013), and to explore how these factors influence judgment in physical tasks.

## 3.2 HEURISTIC MODEL

Prior studies often employ predefined heuristics to elucidate human biases (Schwartz and Black, 1999; Kozhevnikov and Hegarty, 2001; Smith et al., 2017) or fit heuristic models on human data to evaluate the influence of physical attributes (Gerstenberg et al., 2017). While these approaches offer insights for specific tasks, a systematic methodology for learning heuristics in complex scenarios is lacking.

Our heuristic model is designed to learn from a subset of physical attributes, fitting ground-truth data through a direct mapping $g$ from the initial scene $S_0$ to the final judgment $J$, bypassing the intermediate states. This model is advantageous as it approximates humans' real-world physics understanding by a limited set of attributes, and circumvents the need for computationally heavy physics simulation. In particular, we employ a linear model with learnable parameters:

$$J = g(S_0^1, ..., S_0^n) = \sum_{i=1}^{n} \omega_i S_0^i + b, \tag{2}$$

where $\{S_0^i\}$ are different physical variables in $S_0$ and we set $n = 4$ in our study. Specifically, the model considers the following four variables: object size, filling height, object shape, and cup shape. Instead of directly predicting the pouring angle, the model predicts the difference between the actual pouring angle and a reference 90-degree angle. This design choice was made based on preliminary observation from a familiarization experiment that an H-shape cup containing little marbles almost always pours out at 90 degrees. The model is optimized using the mean squared error. Future exploration may consider nonlinear heuristic models using symbolic regression (Xu et al., 2021).

## 3.3 DUAL-PROCESS MODEL

Building on the notion that human cognition might employ multiple systems (Kahneman, 2011), we introduce a dual-process model in the context of intuitive physics, termed SHM. This model hypothesizes that humans alternate between two strategies—mental simulation and heuristic reasoning—based on the duration of the task. Specifically, for duration time below a critical boundary $\theta$, IPE is favored, whereas beyond $\theta$, a heuristic strategy is triggered. This adaptive approach is formalized as:

$$\begin{cases} J = E_\epsilon[M(S_0; \epsilon)], & \text{if } T \leq \theta, \\ J = \sum_{i=1}^{n} \omega_i S_0^i + b, & \text{if } T > \theta \end{cases}. \tag{3}$$

where we drop the dependency on $f$, which remains constant across the same set of experiments. We employ a grid search method to optimize both $\theta$ for the strategic transition and the noise parameters $\sigma$ for the IPE, in addition to a group of heuristic parameters $\omega$ derived from linear regression.

## 4 EXPERIMENT

### 4.1 PARTICIPANTS

A total of 43 college students (55% male, 45% female; mean age = 21.77 ± 4.45) were recruited to participate in in-person experiments. Participants were compensated either with course credits or monetary rewards. One participant was excluded from the analysis due to minimal variation in their responses. It's important to note that we have obtained IRB approval from the Committee for Ethics and the Protection of Human and Animal Welfare at the local institution.

### 4.2 STIMULI

Stimuli were generated using Pymunk in various configurations, encompassing three cup shapes (H-shape, A-shape, and V-shape), three object shapes (circle, triangle, and trapezoid), three object sizes (large, medium, and small), and two filling heights (half and full), totaling up to 54 different conditions. These stimuli were then rendered using Pygame.

In each condition, marbles were randomly placed inside cups, and their layouts were automatically adjusted by Pymunk's physics engine to create physically plausible scenarios. For each condition, three random layouts were generated. The selection of images as stimuli occurred when the marbles reached a stable state. The marbles were designed to have neither friction nor elasticity and equal mass since the estimation of those variables brings additional costs for participants and disturbs the results. The marbles were assigned colors randomly from a grayscale palette to eliminate any prior knowledge of material properties.

Pouring angles for each trial were determined through controlled simulations, with cups undergoing slow rotations. The angle at which marbles began falling out—identified when a marble's mass center aligned with the cup's top-left corner—was measured. This measurement involved calculating each marble's dynamics at an FPS of 120 and automatically detecting the falling-out event. The tilt angles, indicating changes in the central axis of the cups, were referenced to ensure accuracy in each trial's pouring angles. Example stimuli are presented in fig. 2a.

### 4.3 PROCEDURES

A within-subjects design was implemented, where each participant completed all 54 conditions. Stimuli were navigated in a counter-balanced order with randomly selected layouts, and the experiment lasted approximately 30 minutes.

**Familiarization** After completing a consent form, participants were asked to read instructions and complete a familiarization session involving videos of pouring two small marbles; see appendix A and appendix B for details. This session is aimed at familiarizing participants with (i) the properties of marbles and their physical dynamics, (ii) the definition of the tilt angle, and (iii) the concept of "pouring out." Quizzes were conducted after each concept familiarization to ensure the participant's full understanding. The first quiz required participants to determine the angle of two tilted empty cups. The following quiz asked participants to select the moment when marbles would pour out from cups in three different scenarios. Only upon passing these quizzes were participants permitted to proceed to the next experiment phase.

**Experiment** Participants were required to complete 54 trials consecutively. In each trial, a static image of a non-rotated cup from various setups was presented. The tilt angle necessary for the cup to begin pouring out marbles was estimated by the participants using a slider bar with a range of 0 to 135 degrees. To reduce potential biases from inaccurate angle perception, a dial marked with angle measurements was provided in each trial. Demographic information along with the responses for the pouring angles across all 54 trials, including the total duration, were recorded for subsequent analysis.

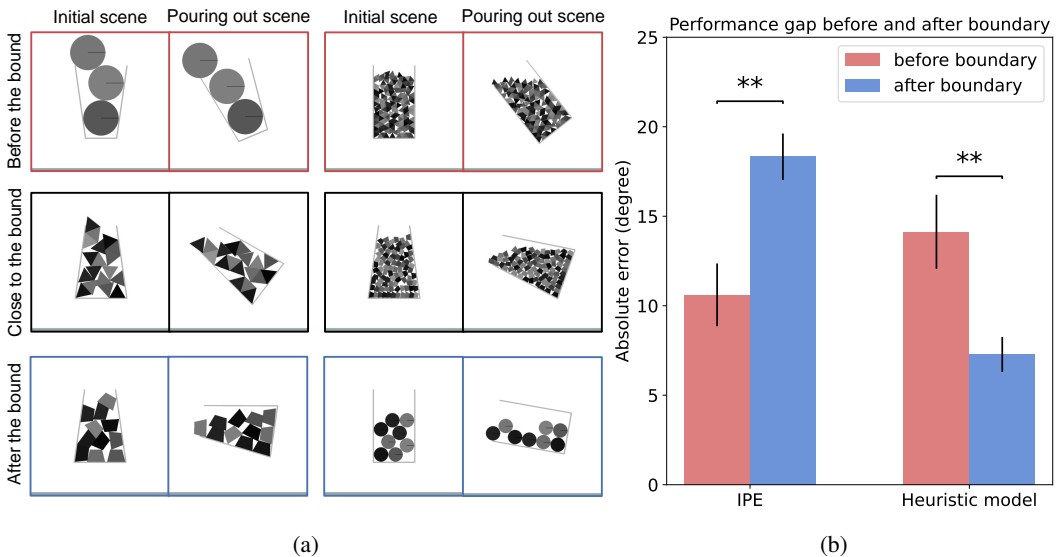

Figure 2: Visualizations of stimuli and error analysis. (a) Example stimuli. The top (red), middle (black), and bottom (blue) rows depict two scenarios each, with pouring angles that are smaller, close to, and larger than the established simulation bound, respectively. (b) The mean absolute error between model and human results (with SEM). The IPE model exhibits a larger absolute error when the simulation time exceeds the boundary. Conversely, the heuristic model shows contrary results, indicating its effectiveness in these scenarios.

**Feedback** A feedback session was held post-experiment to gather participants' comments, particularly focusing on the strategies employed during the task.

## 5 RESULTS

In this section, we follow a four-step approach to validate our hypothesis. First, we analyze participants' error patterns, which suggest a shift in reasoning strategies. Second, we test the IPE to account for human judgment, but it falls short in explaining the observed underestimation pattern. Third, we enhance the IPE by integrating a heuristic model that considers key physical attributes. Finally, in the fourth step, we develop a hybrid model that incorporates both simulation and heuristic models, using a switching mechanism to best explain human judgments across all conditions.

### 5.1 A SWITCHING IN ERROR PATTERNS

Human results show overestimation and underestimation of the pouring angle compared with the ground truth. These two error patterns may indicate different strategies of physical reasoning. To examine whether there is a switching mechanism between the two patterns among those conditions, we employed symbolic regression to automatically identify an explainable factor and its corresponding switching point that best distinguishes between the two patterns. We considered all experimental design factors, including cup shape, object shape, object size, and filling height, along with the object number and simulated pouring angle. Our analysis shows that the simulated pouring angle effectively differentiates between the reversal patterns observed in human participants' estimations of tilt angles for pouring (see fig. 1C). We identified the optimal boundary for distinguishing these patterns to be 65 degrees by searching from 20 to 120 with an interval of 1. Initially, participants tended to overestimate these angles when the simulated pouring angles were relatively small (mean discrepancy = 7.76 ± 13.67). As the angle increases, this trend shifts to consistent underestimation (mean discrepancy = -9.89 ± 8.75). Given the consistent tilting speed, the observed pattern switch as the pouring angle increases suggests a hypothesis that the physical reasoning strategy may change when the simulation time exceeds a certain resource boundary.

## 5.2 IPE FAILS TO EXPLAIN ALL TRIALS

To validate our hypothesis, we first experiment with the IPE model. Fitting human judgments in the overestimation phase with IPE supports our hypothesis of the simulation strategy's dominance in the shorter time span. Note that as the angular speed remains constant in our experiments, the simulation time is proportional to the degree of angle. When the positional noise and rotational speeds of the IPE model were optimized (see section 3.1 for details), the results were closely aligned with human performance, explaining the overestimation pattern effectively (r = .890).

However, once the pouring angle exceeded the 65-degree boundary, IPE's prediction error significantly increased (t(52) = -3.354, p = .002; see fig. 2b for absolute error comparison on the left). No parameter combination in the IPE model could well explain the underestimation pattern, indicating the existence of an alternative strategy other than IPE. See visualization of IPE simulation in appendix C.

## 5.3 LEARNED HEURISTIC MODEL COMPLEMENTS IPE

To better explain the underestimation pattern in human behavior, we devised a heuristic model incorporating key physical attributes rooted in our experiments: filling height, cup shape, object shape, and object size. This model effectively compensated for the discrepancies unexplained by the IPE model. The heuristic model performed well when the actual pouring angle exceeded 65 degrees (r = .841), but its accuracy diminished below this boundary (Mann-Whitney U test, p = .003; see fig. 2b for absolute error comparison on the right).

Further analysis of specific heuristics revealed that filling height, cup shape, and object size significantly influence heuristic judgment (see table 1, p = .000 for all three variables). The model's coefficients allowed a quantitative assessment of these variables' impact. For example, V-shaped cups, with outwardly sloping walls, require smaller tilt angles for pouring, typically 11.528 degrees earlier than H-shaped cups. Larger marbles increased the tilt angle required for pouring by 7.029 degrees compared to smaller ones. Cups filled to a higher level poured out earlier, 19.955 degrees less than half-filled ones on average. Despite the simplicity and approximate encoding, this linear heuristic model captured basic physical intuition effectively. The findings align with our second hypothesis, suggesting the adoption of heuristic strategies when mental simulation reaches its boundary.

Table 1: **Categories, coefficients, and p-values of physical variables in the learned heuristic model.** All physical variables except the object shape show significant contributions to the outcomes.

| Variable | Category | Coefficients | p |
|---|---|---|---|
| Cup shape | H-shape | -11.528 | 0.000 |
| | A-shape | | |
| | V-shape | | |
| Object shape | Circle | 1.577 | 0.073 |
| | Triangle | | |
| | Trapezoid | | |
| Object size | Small | 7.029 | 0.000 |
| | Medium | | |
| | Large | | |
| Filling height | Half | -19.955 | 0.000 |
| | Full | | |

## 5.4 SHM EXPLAINS HUMAN JUDGMENTS ON ALL CONDITIONS

Building upon our findings, we constructed the Simulation-Heuristics Model (SHM), a dual-process model integrating both simulation and heuristic strategies, to optimally predict human performance across all trials. Instead of relying on actual simulation time in humans, which is unavailable, we instead based the transition criterion in SHM on IPE's simulation time. A grid search identified the boundary of 68.2 degrees in simulation time and a dynamic positional noise of 0.2 as optimal for mirroring human judgments.

In predicting overall human performance, SHM surpassed three baseline models: the deterministic physics model, IPE, and the purely heuristic model. SHM exhibited the highest correlation and lowest RMSE (r = .834, RMSE = 10.002), as shown in fig. 3. Although IPE was correlated with human judgments (r = .772), it showed high error in making human-like predictions (RMSE =

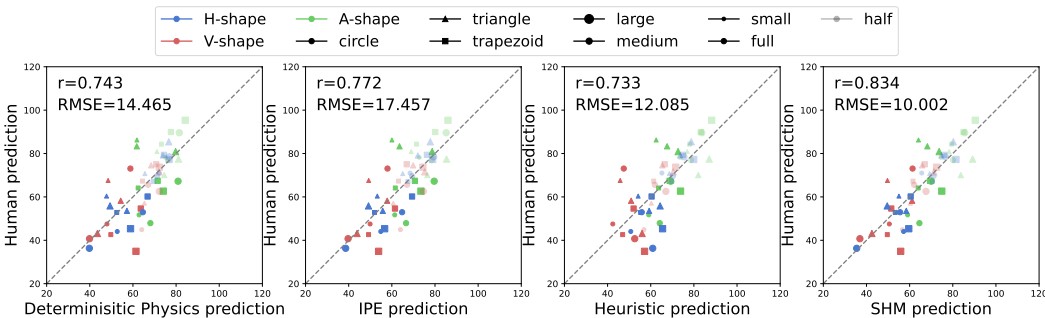

Figure 3: **Comparison between SHM and other baseline models.** The correlation and RMSE between model predictions and human predictions across all 54 conditions are compared. Among the four models evaluated, SHM demonstrates the highest correlation and the lowest RMSE, indicating its superior predictive accuracy.

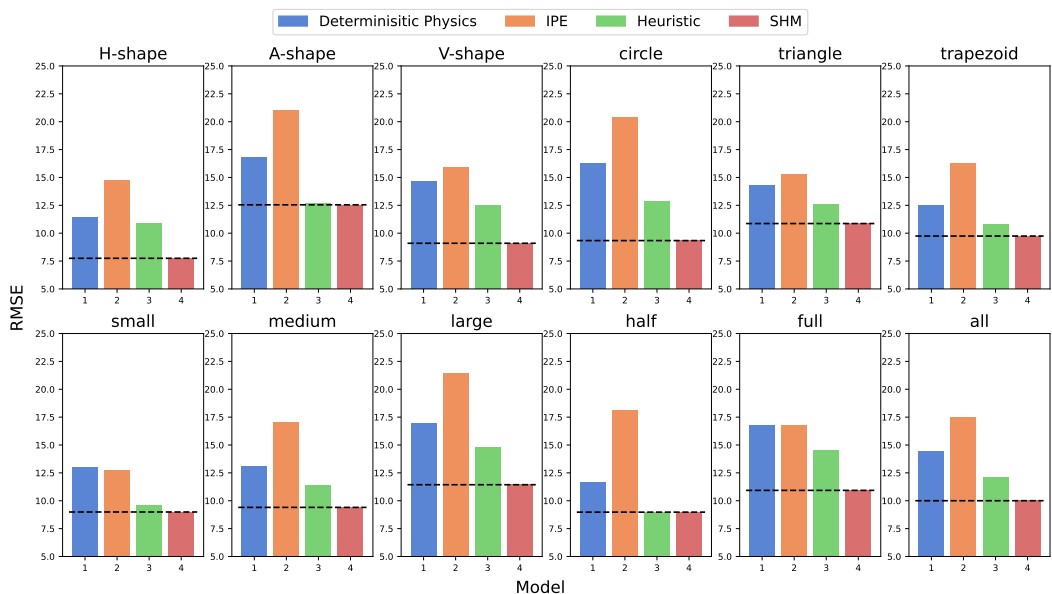

Figure 4: **Comparison of four models' RMSE on different conditions.** RMSE is calculated as the root mean square error between the model's predicted pouring angle and the human judgments. The bottom right figure represents the performance across all 54 trials. A dashed line is included to indicate the RMSE of the SHM, showing a clear advantage when compared with other models.

17.457). On the contrary, the heuristic model could predict human judgments with smaller RMSE but failed to better explain the variance (r = .733, RMSE = 12.085).

The fitted SHM model exhibited strong generalization across various scenarios (e.g., different cup shapes, object shapes, sizes, and filling heights). It consistently showed the lowest RMSE, except in specific scenarios where the heuristic model was parallel (fig. 4). The model explained maximum variance in almost all cases, with comparable performance to IPE in scenarios involving large or trapezoidal marbles. Notably, in situations where IPE minimally correlated with human judgments (e.g., A-shaped cups, r = .461), SHM maintained effectiveness (A-shaped cups, r = .647). It also significantly improved correlation in scenarios poorly addressed by the heuristic model (full filling height, r improved to .673 from .377). These results highlight SHM's capability to synergize the strengths of both IPE and the heuristic model, enabling robust predictions across diverse scenarios. Consequently, the SHM model, with its transition mechanism based on simulation time, aligns with our third hypothesis and effectively accounts for a wide range of conditions and metrics.

## 6 DISCUSSION

**Contributions** The novelty of our study resides in its quantitative analysis of the simulation and heuristic strategies and their transition mechanism. We introduced and validated the SHM model, which illuminates an intriguing pattern: as simulation cost increases (indicated by simulation time in our experiments), the cognitive strategy shifts from detailed simulation to more generalized heuristic reasoning based on key physical attributes. These heuristic methods, despite biased, facilitate quick and reasonably accurate judgments in complex scenarios. Our pouring task results show that the heuristic model leans towards slight underestimation, a conservative strategy that potentially ensures safety in execution. The SHM model's efficacy is highlighted by its improved correlation with human judgments and reduced error rates compared to models relying solely on simulation or heuristic approaches. These results underline the significance of a hybrid model in capturing human cognitive processes, especially in intuitive physics. Apart from offering insights into human cognitive adaptability, our findings have implications for advancing computational models in artificial intelligence, integrating dynamic prediction capabilities with heuristics-based reasoning for a more nuanced understanding of the physical world.

**Limitations** Our study suggests a possible method for distinguishing between two cognitive strategies. However, it is crucial to recognize that individuals may not consistently adhere to one strategy during a trial. This is particularly true in complex situations where transitions between strategies may occur during the decision-making process. Investigating these transitions within a single scenario poses significant challenges. The main difficulty arises in detecting and interpreting intermediate signals that may not display distinct patterns, complicating the analysis compared to identifying the primary strategy that impacts final decisions. Furthermore, certain outlier cases remain unexplained by existing models, indicating a need for more nuanced modeling to better capture the variability in human judgment.

**Future work** Future research could expand upon our findings by exploring various physical scenarios. While the use of pouring angles as a transition metric is specific to our task, the underlying transition mechanism based on simulation time might have broader applicability across various contexts. Additionally, incorporating nuanced cognitive factors could deepen our understanding of intuitive physics. Although simulation time emerged as a significant predictor of human judgment, we observed improved model performance when considering shifts in simulation time influenced by cup shapes. This suggests that other criteria, such as the complexity of the simulation process or scenario familiarity, might also play crucial roles. Future studies could aim to quantify these aspects to better explain transitions in intuitive physics strategies.

In different scenarios, the dual-process model used by humans may utilize different physical variables as heuristics. For example, when estimating the collapse of a block tower, the height of the tower might serve as a heuristic, while in predicting the motion of a group of balls, an approximate distribution of the balls' positions could be used as a heuristic. Despite these variances and the diversity of heuristic strategies, it would be intriguing to explore whether our proposed learning approach remains effective across different contexts.

## 7 CONCLUSION

In this work, we design a pouring-marble task to study the computational mechanism in intuitive physics. The sequential experiments underscore that while the IPE effectively predicts human judgments in scenarios with short simulation times, its efficacy diminishes as these times extend. This limitation of IPE paves the way for the implementation of a heuristic approach that shows greater accuracy in scenarios necessitating longer simulations. The introduction of the SHM model, which integrates these two cognitive strategies based on the simulation cost of the task as approximated by simulation time, not only aligns more closely with human behavior but also enhances the model's generalization capabilities across varied conditions. By bridging the gap between mental simulation and heuristic approaches, the SHM model offers a robust framework that captures the complexity and adaptability of human cognition in intuitive physics. This model serves as a pivotal step in exploring computational methods that mimic human-like reasoning, providing insights into the cognitive mechanisms that govern our interactions with the physical world.

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

## A    INSTRUCTIONS TO PARTICIPANTS

Your task is to judge the angle at which a tilted container will begin to pour out its contents. Please read the following instructions carefully.

- In each trial of this experiment, you will be given an image of a container filled with objects.
- The objects have no friction and no elasticity.
- The masses of objects are the same although their colors may be different.
- The container will be tilted slowly to the left at a very slow speed (!!!), and objects will start to pour out at a certain angle (the tilting angle indicates the change of the central axis of the container).
- However, you will only be given the static image of the initial scene rather than the whole tilting process.
- The pouring-out time is the moment when the center of mass of any object exceeds the top-left corner of the cup.
- Your judgment should be as accurate as possible.
- For your reference, you can also see an image of visualized angles and drag it to the cup with your mouse to measure the angle:

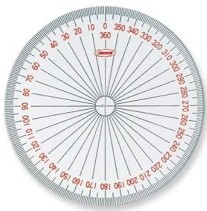

Here are three videos to help you familiarize yourself with the settings.

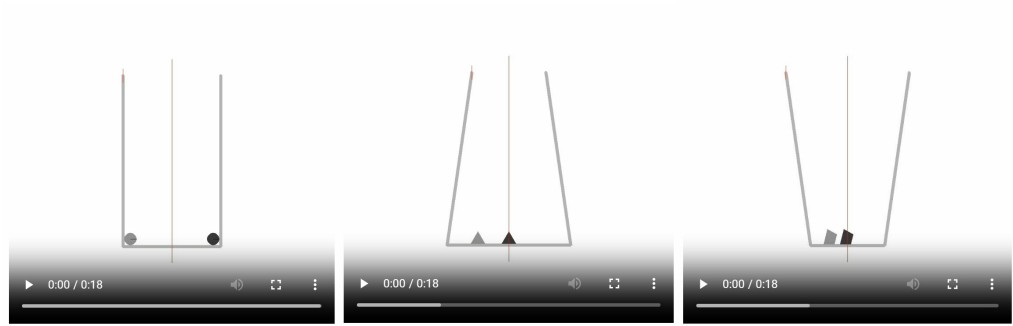

Please note that:

- You are only allowed to play each video two times at most.
- Pay attention to how objects move and the pouring out moment.
- The video is accelerated to save you time.

## B    SCREENSHOTS

We present the screenshots of our two familiarization quizzes. The first quiz assesses their understanding of tilting angles as shown in fig. A1, while the second quiz focuses on their grasp of the pouring out concept as shown in fig. A2.

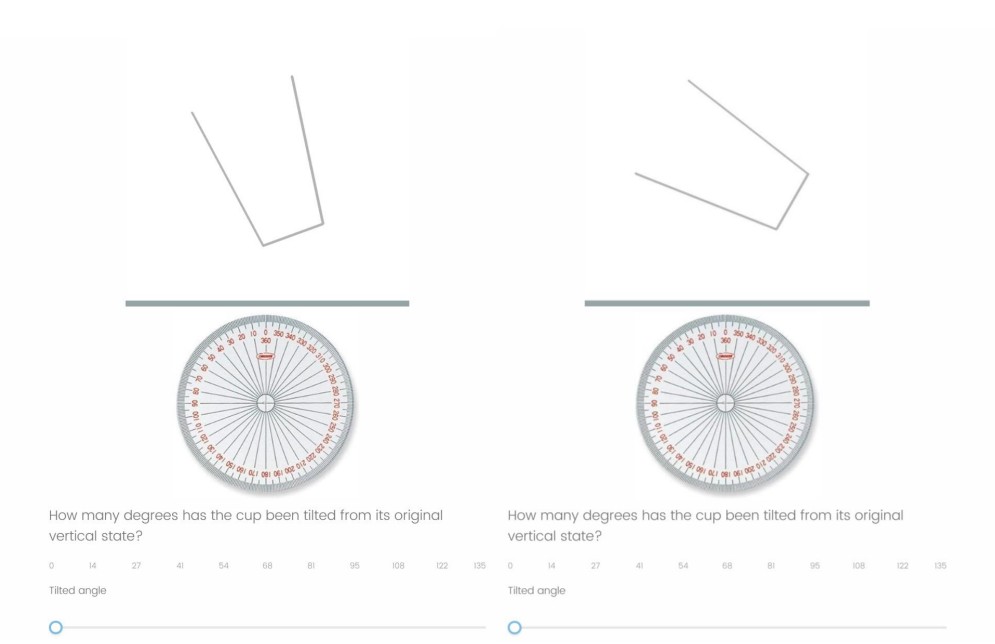

Figure A1: **Screenshot of the familiarization quiz about titling angle.**

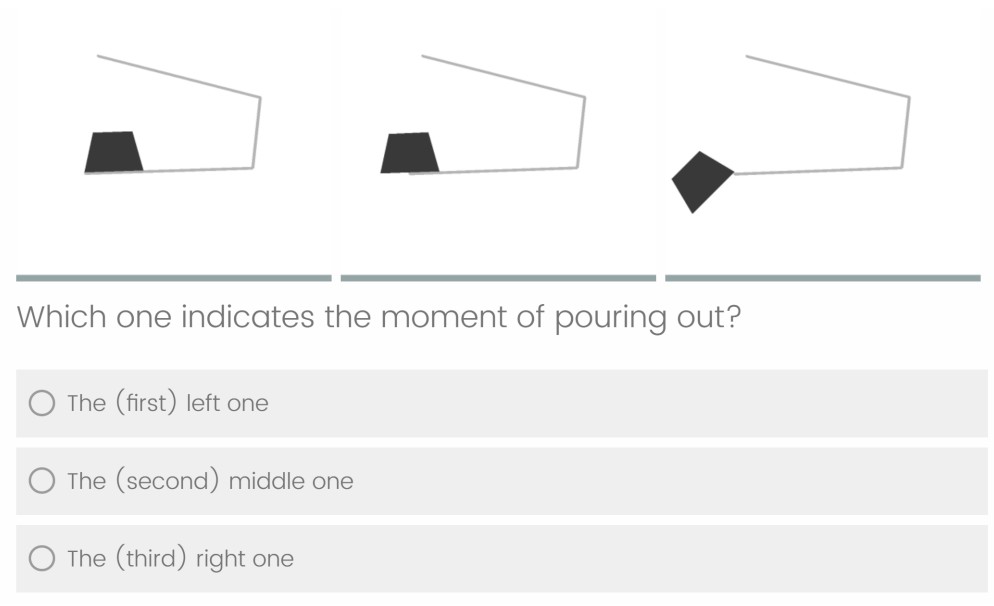

Figure A2: **Screenshot of the familiarization quiz about pouring out moment.**

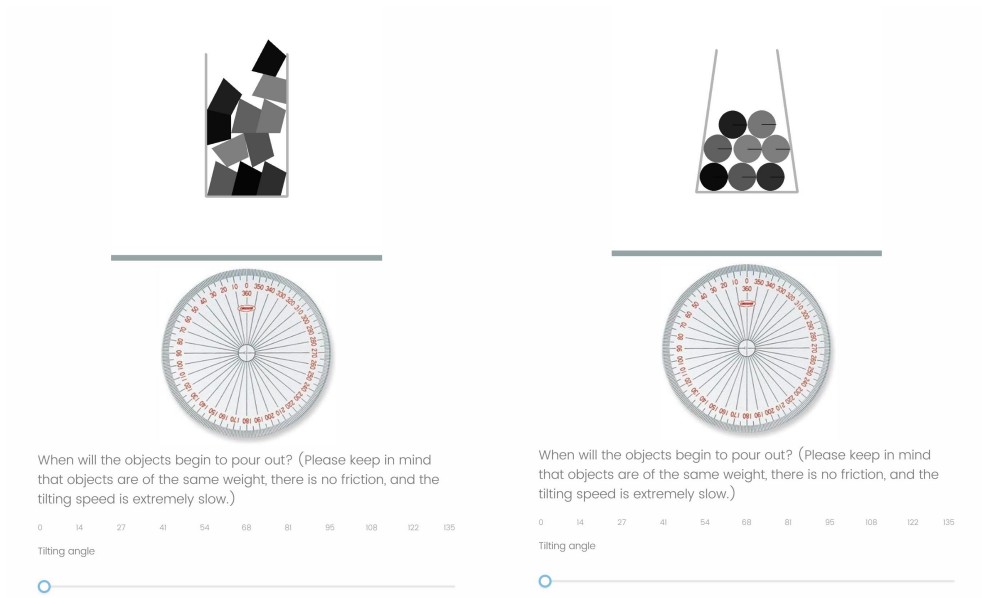

Figure A3: **Stimuli examples.**

## C  VISUALIZATION

We present an example of IPE simulation results in fig. A4 to show how different noise perturbations can affect the physical dynamics in our pouring-marble task.

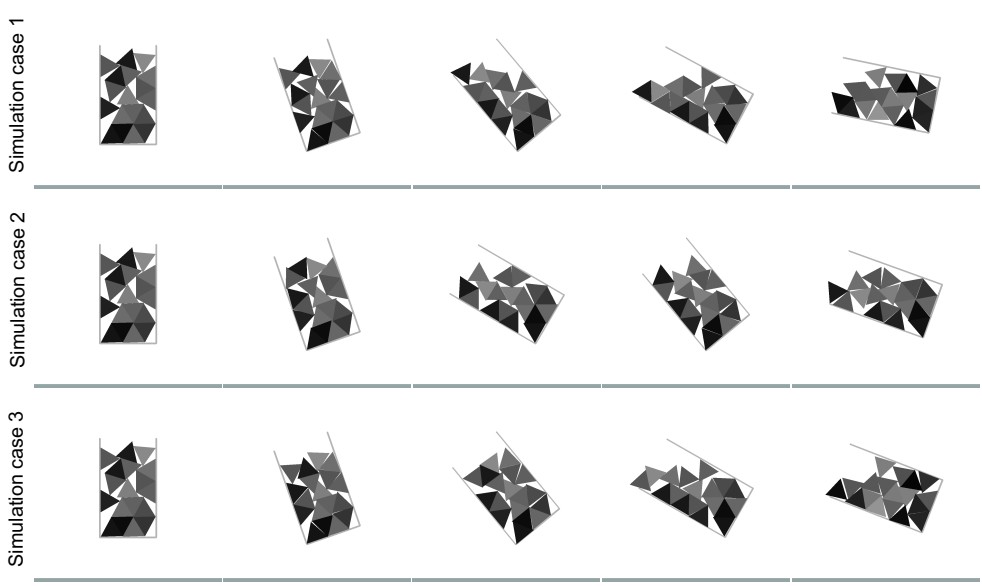

Figure A4: **Visualization of IPE simulation.** The shown condition includes a regular cup fully filled with medium-sized triangle marbles. The three cases show the dynamics altered by different noise perturbations from a specified distribution.

