# OpenReview forum: "A simulation-heuristics dual-process model for intuitive physics"
_ICLR.cc/2025/Conference — ICLR 2025 Conference Withdrawn Submission_

### Official Review · Reviewer_YZE6 · 2024-10-17

**Soundness:** 1
**Presentation:** 3
**Contribution:** 2
**Rating:** 3
**Confidence:** 3

**Summary:**

This work investigates the role of mental simulation and heuristics in human physics prediction. Whereas previous works have studied mental simulation and heuristic-driven physics prediction in isolation, this paper hypothesizes that humans make use of both of these strategies and switch between them depending on the context and problem difficulty. They design a new “pouring marble task” with more diverse physical properties. Humans are asked to judge the tilt angle needed to pour marbles from cups under various setups.

**Strengths:**

Overall, the paper was well-written and easy to understand. The argument was very clearly laid out in the introduction, and the figures look nice.

**Weaknesses:**

- When I first read “We employ a grid search method to optimize both θ for the strategic transition and the noise parameters σ for the IPE, in addition to a group of heuristic parameters ω derived from linear regression.”, I wasn’t quite sure what that meant. What was the objective being maximized? Further on in the paper, the authors state “A grid search identified the boundary of 68.2 degrees in simulation time and a dynamic positional noise of 0.2 as optimal for mirroring human judgments." Does this mean that the Simulation-Heuristics Model is fit to the human data? If so, are there separate cohorts for model fitting and model comparison?

- I don’t understand the newly proposed heuristic model. As stated in the paper, previous work in heuristic models use predefined rules or fit to human data. In this paper, the heuristic model is fit to simulation data. How is this an accurate representation of human heuristics since humans haven’t seen the examples this model has.

- Considering this was only tested on a single problem (pouring), it seems premature to claim this as a general model for resource efficient physics prediction. If correct, we would see this behavior replicated across a variety of tasks using some unified notion of a computational budget. Time is also not a very good proxy for simulation difficulty.

- The authors report correlations and RMSE for different models (Heuristic, IPE, SHM), but I couldn't find any statistical tests that compared these models.

- Would the findings in this paper not be consistent with a biased heuristic or simulation model? Is there any reason a miscalibrated physics simulation that got the friction, mass, etc parameters wrong wouldn’t also lead to angle-dependent prediction error?

- In figure 1c, red is used for both the mean angle estimate and the A-shape points. This was confusing at first glance.

- No code is provided, which limits reproducibility.

**Questions:**

See weaknesses

---

### Official Review · Reviewer_fRVk · 2024-11-04

**Soundness:** 2
**Presentation:** 2
**Contribution:** 1
**Rating:** 3
**Confidence:** 4

**Summary:**

This paper proposes a new framework - Simulation Heuristic Model (SHM) which is built on top of a linear heuristic model to replicate human prediction as opposed to time-expensive simulation.

**Strengths:**

The paper asks original questions along the lines of how humans reason and how often they employ simulations vs heuristics to make predictions.
The paper performs a user study to understand the impact and significance of the idea.

**Weaknesses:**

The paper is poorly written and presented. Contributions and results - which should be highlighted and form the thesis of the paper - are buried in details.
If I understand correctly, the primary idea in the paper is to approximate a noisy simulation (which is computed using equation 1) using a linear model (represented in equation 2) after a certain time threshold for the simulation is met. There are two issues here - I don't think physical simulations can be represented using a quartic equation. Were other heuristic models considered? Additionally, deciding on the time threshold after which the heuristic should kick in requires having a hold out validation set. How was this done with having only 43 participants?

**Questions:**

1. How did other heuristic models do (for example neural nets)? What are the inputs into this heuristic model?
2. Are results from a user study of 43 participants statistically significant to claim that this dual process works better than IPE? Was a hold out validation set used for computing results using a time threshold deciding by grid search on a training set?
3. If the heuristic (equation 2) is supposed to "mimic/predict" the simulation (equation 1), how is it possible for SHM to outperform IPE in practice? What data was the heuristic trained with?

---

### Official Review · Reviewer_EBaN · 2024-11-07

**Soundness:** 3
**Presentation:** 4
**Contribution:** 3
**Rating:** 8
**Confidence:** 2

**Summary:**

This work presents a methodology design for looking into intuitive physics engine. A pouring-marble task is designed with various conditions and the results show some interesting behavior in cognitive strategies. Inspired by this, a framework called SHM is proposed for human mental simulation that aligns more precisely with human behavior.

**Strengths:**

The research topic is interesting and, compared with previous work, the scenario is more complicated and the experiment shows the effectiveness of the new modeling approach.

**Weaknesses:**

An important contribution claimed in this paper is that, compared with previous works that mainly focused on a single task, this work provides a systematic methodology for learning heuristics. However, there is only one task in this paper although with varied conditions. I recommend adding another task with similar settings to show the general utility.

**Questions:**

The modeling approach aims for a systematic methodology. How general is this model? Can this model handle some scenarios that the boundary cannot be described with a single parameter?

---

### Official Review · Reviewer_u9Qr · 2024-11-08

**Soundness:** 2
**Presentation:** 2
**Contribution:** 2
**Rating:** 3
**Confidence:** 3

**Summary:**

The paper introduces a new framework, the Simulation-Heuristics Model (SHM), which conceptualizes intuitive physics as a dual process: Intuitive Physics Engine (IPE) dominates in short-term simulations, while a heuristic-based approach takes over when the IPE’s simulation extends beyond a certain time boundary.

**Strengths:**

The paper introduces a new framework, the Simulation-Heuristics Model (SHM), which conceptualizes intuitive physics as a dual process: Intuitive Physics Engine (IPE) dominates in short-term simulations, while a heuristic-based approach takes over when the IPE’s simulation extends beyond a certain time boundary.

**Weaknesses:**

-I don't think this work is suitable for submission to ICLR, as it lacks AI/ML elements, learning representation and primarily consists of human experiments. I would recommend the author consider submitting it to CogSci or another more relevant conference.

-Too simple task scenarios, it would more convincing to see how this SHM can helped with other downstream real-world tasks?

-How well can existing VLM do in the proposed tasks?

**Questions:**

refer to weakness.

---

### Note · Authors · 2024-12-01

I have read and agree with the venue's withdrawal policy on behalf of myself and my co-authors.